# Perinatal care in Western Uganda: Prevalence and factors associated with appropriate care among women attending three district hospitals

Mercy Muwema[1]*, Dan K. Kaye[2], Grace Edwards[3], Gorrette Nalwadda[4], Joanita Nangendo[1], Jaffer Okiring[1], Wilson Mwanja[5], Elizabeth N. Ekong[6], Joan N. Kalyango[1,7], Joaniter I. Nankabirwa[1,8]

1 Clinical Epidemiology Unit, School of Medicine, Makerere University College of Health Sciences, Kampala, Uganda, 2 Department of Obstetrics and Gynecology, School of Medicine, Makerere University College of Health Sciences, Kampala, Uganda, 3 School of Nursing and Midwifery, Aga Khan University, Kampala, Uganda, 4 Department of Nursing, School of Health Sciences, Makerere University College of Health Sciences, Kampala, Uganda, 5 Whale Consult Limited, Kampala, Uganda, 6 Department of Nursing, Faculty of Health Sciences, Uganda Christian University, Uganda, 7 Department of Pharmacy, School of Health Sciences, Makerere University College of Health Sciences, Kampala, Uganda, 8 Infectious Diseases Research Collaboration, Kampala, Uganda

* muwecy@gmail.com

**Data Availability Statement:** All relevant data are within the manuscript and its Supporting Information files.

## Abstract

### Background

Perinatal mortality remains high globally and remains an important indicator of the quality of a health care system. To reduce this mortality, it is important to provide the recommended care during the perinatal period. We assessed the prevalence and factors associated with appropriate perinatal care (antenatal, intrapartum, and postpartum) in Bunyoro region, Uganda. Results from this study provide valuable information on the perinatal care services and highlight areas of improvement for better perinatal outcomes.

### Methods

A cross sectional survey was conducted among postpartum mothers attending care at three district hospitals in Bunyoro. Following consent, a questionnaire was administered to capture the participants' demographics and data on care received was extracted from their antenatal, labour, delivery, and postpartum records using a pre-tested structured tool. The care received by women was assessed against the standard protocol established by World Health Organization (WHO). Poisson regression with robust standard errors was used to assess factors associated with appropriate postpartum care.

### Results

A total of 872 mothers receiving care at the participating hospitals between March and June 2020 were enrolled in the study. The mean age of the mothers was 25 years (SD = 5.95). None of the mothers received appropriate antenatal or intrapartum care, and only 3.8% of

**Funding:** The author(s) received no specific funding for this work.

**Competing interests:** The authors have declared that no competing interests exist.

the participants received appropriate postpartum care. Factors significantly associated with appropriate postpartum care included mothers being >35 years of age (adjusted prevalence ratio [aPR] = 11.9, 95% confidence interval [CI] 2.8–51.4) and parity, with low parity (2–3) and multiparous (>3) mothers less likely to receive appropriate care than prime gravidas (aPR = 0.3, 95% CI 0.1–0.9 and aPR = 0.3, 95% CI 0.1–0.8 respectively).

## Conclusions

Antenatal, intrapartum, and postpartum care received by mothers in this region remains below the standard recommended by WHO, and innovative strategies across the continuum of perinatal care need to be devised to prevent mortality among the mothers. The quality of care also needs to be balanced for all mothers irrespective of the age and parity.

## Introduction

Improvement in the quality of perinatal care is essential in reducing maternal and neonatal mortality, and is required in order to meet the Sustainable Development Goal three target aimed at: 1) reducing the maternal mortality ratio (MMR) to less than 70 deaths/100,000 live births by 2030; 2) reducing the neonatal mortality rate (NMR) to less than 7/1000 live births by 2035; and 3) reducing the number of still births to less than 8 still births/1000 total births [1–3]. Sub Saharan Africa bears the highest burden of adverse maternal and neonatal outcomes, contributing 55% of the global stillbirths [4], 66% of the global maternal mortality [5], and 46% of the neonatal mortality [6]. In these same settings, countries also struggle with achieving the set standards of quality care for pregnant women [3].

Uganda is one of the countries that face challenges in provision of quality perinatal care [7–11]. According to the latest 2016 Uganda Demographic and Health Survey (DHS), only 60% of the women surveyed had attended the recommended four antenatal visits during the pregnancy leading to their most recent birth, and less than one third had their first visit during the first trimester of pregnancy [7]. In addition, only 39% of the women had had a urine sample taken, and even though 74% of childbirths were attended by skilled health personnel, only 54% of the women and 56% of the newborns received a postnatal check within 2 days of delivery. In an independent study evaluating antenatal care among 299 mothers in eastern Uganda 53% of the mothers did not have essential tests conducted, 62% were not offered sulphadoxine pyrimethamine (SP) for intermittent treatment of malaria (IPTp), and 72% were not offered folic/ferrous sulphate supplement [8]. Similar findings were reported in other studies across the country [9–11]. Furthermore, education, occupation, age, parity, being wealthy and caesarian section as a mode of delivery were found to be associated with perinatal care [12–14].

Bunyoro, a region in western Uganda is being supported by a number of quality improvement projects in order to reduce the risk of maternal and infant mortality including: 1) the Clinton Health Access Initiative (CHAI) that has continued to equip and build the capacity of the health system to ably manage, deliver and sustain improvements in reproductive, maternal, new born, child and adolescent health programming since 2007 [15]; 2) the Ministry of Health reproductive, maternal, and child health improvement project (URMCHIP) which aims at improving the utilization of essential reproductive, maternal, newborn, child and adolescent health services [16]; and 3) Saving Mothers Giving Life project (SMGL), a health system strengthening project that ran from 2012 to 2014. The SMGL's aim was to address the three

delays in health care including the delay in decision to seek care, in reaching care, and in receiving adequate and appropriate health care [17]. Despite this support, the region registered the nation's lowest scores on several indicators of maternal health care in the 2016 DHS survey with the number of mothers attending the recommended four antenatal visits at 44.5%, proportion of women who had a urine sample examined during antenatal visits at 27.5%, and proportion of women who had no post-natal checks within 48 hours of delivery at 63.8% [7].

Since the last national evaluation of maternal care as part of the 2016 DHS survey, the World Health Organization (WHO) has updated its guidelines on maternal health care [18]. These new WHO guidelines were adopted and rolled out in Uganda in 2018 [19], however, no survey has evaluated the quality of care especially in the underperforming regions of the country since their role out. This study assessed the prevalence and factors associated with appropriate perinatal care at the three district hospitals in Bunyoro region. The study findings document the current quality of maternal care in the region following the support received from the different implementing partners (URMCHIP and CHAI) and change in the national guidelines.

## Materials and methods

### Study design and setting

A facility based cross-sectional study was conducted between March and June 2020 in three public district hospitals of Bunyoro region, Uganda. The region is comprised of eight districts that include Kakumiro, Kibaale, Kagadi, Kikuube, Hoima, Masindi, Buliisa, and Kiryandongo [20]. The region's population was estimated at 2,028,500 million people in the 2014 National Demographic and Population Census [21]. Bunyoro region has one of the highest fertility rates (7.5) in the country, and has a high proportion of both teenage pregnancies (10.6%) and early marriages (19%) [22]. The region has three district hospitals (Kagadi, Kiryandongo and Masindi). A district hospital is the highest-level public health facility in any given district and covers a catchment population of approximately 500,000 people. The district hospital offers preventive, promotive, and both in and out patient curative services in all areas of child and adult medicine [23]. It is also responsible for supervising and planning for all the lower-level facilities within the district. An average of 300 deliveries are registered at these hospitals in any given month.

### Study population

Postpartum mothers in the three participating district hospitals were screened at discharge for eligibility to join the study. A mother was eligible for inclusion if: 1) she attended antenatal care in the study hospitals; 2) she gave birth in the study hospitals; 3) delivery was conducted by a skilled health professional; 4) she provided written informed consent to participate in the study; and 5) she had a health record indicating care received during the antenatal, intrapartum and postpartum periods.

### Sample size and sampling

We hypothesized that the quality of care would be different between educated and less educated mothers and therefore computed sample size using formula for comparison of two proportions. The estimated proportions among less educated (64.1%) and educated women (24.7%) were based on a study done in Nepal [24]. We further assumed 5% level of significance, 80% power, design effect of 2 to cater for clustering at health facility level, and nonresponse of 10% which gave an estimated sample size of 755 mothers. Using probability

sampling proportionate to size, we determined the mothers to be enrolled from each hospital and used consecutive sampling within each of the hospitals.

## Study variables

The variables of the study included antenatal care, intrapartum care, and postpartum care. Antenatal care was assessed as the number of antenatal contacts, initiation of antenatal care, examinations during every contact, tests conducted, preventive drugs, ultrasound scan, tetanus toxoid, and health education. The intrapartum care aspects assessed included use of a partograph, monitoring the fetal heart rate, monitoring progress of labour (cervical dilatation, descent of presenting part, and uterine contractions), and monitoring of the maternal blood pressure and maternal pulse rates. Finally for postpartum care, the monitoring of the uterine contraction, vaginal bleeding, fundal height, blood pressure, pulse rate, temperature, and urine voiding were assessed as the parameters of care.

## Data collection

At each hospital, data was collected by qualified midwives fluent in the local language of the area (Runyoro) and not directly involved in the patient care. Postpartum mothers were identified using the ward registers and screened for eligibility to join the study. Following the consent process, a questionnaire was administered to collect socio-demographic data from participants. Data on care received at the different stages of perinatal care was extracted from their hospital notes using a pre-tested structured data extraction tool designed using the Open Data Kit (ODK) software. Data extracted included the participants' information on preventive medications received, any health education, diagnostic tests and results, and examinations received by the mothers during their pregnancy.

## Data management and analysis

Data collected was downloaded into a CSV file format for cleaning, and exported to STATA version 13 for analysis. Outcomes of interest were assessed using the standard protocols established by WHO recommendations on antenatal care [18], intrapartum care [25], and postpartum care [26]. Care which reached the standard recommended by the WHO was coded as "yes" and given a score of 1 while "no" and score 0 was used for care that did not reach the recommended standards. The standard of care was grouped into three categories: 1) "no care" which was defined as a total care score of zero (the minimum possible score); 2) "appropriate care" which was defined as a total score of 8 for antenatal, 6 for intrapartum care, and 7 for postpartum care (the maximum possible score); and 3) "inappropriate care" which referred to the range of scores between the minimum and maximum total care scores.

Levels of care were summarized and presented as proportions with 95% confidence intervals, stratified by the hospital. Modified poisson regression with robust standard errors was used to assess for factors associated with appropriate postpartum care. At bivariate analysis, all variables that had p-value of less than or equal to 0.20 were considered for multivariate analysis and logical model building was used to generate the final model. Education level and health facilities attended were confounding this model while the mother's income is a known confounder and all three were included in the final model even when not statistically significant. The measures of associations are presented as prevalence ratio (PR) with their 95% confidence intervals and p-values. A p-value of <0.05 is considered significant.

### Ethical considerations

Ethical approval to conduct the study was obtained from the Makerere University School of Medicine Research and Ethics Committee (REC REF# 2019–137) and the Uganda National Council for Science and Technology (HS483ES). Written informed consent to participate in the study was obtained from all participants prior to enrolment in the study, and unique identifiers and not personal names were used for participant identification.

## Results

### Description of the study population

A total of 3,320 women were screened for eligibility to join the study, and 872 (26.3%) were enrolled. The commonest reason for exclusion was having not attended antenatal care at the study hospitals (2,371, 96.9%). Other reasons for exclusion included patients being referred to study hospitals from lower facilities due to intrapartum complications, mothers arriving at the facility after birth (birth before arrival (BBA)) and declining consent to participate in the study. Fig 1 provides details of the participants flow stratified by health facility.

The mean age of the participants was 25 years (SD = 5.95). Majority of the participants were married or in a stable relationship (n = 782, 89.7%), and more than half (n = 453, 52%) had never received any education. Although many of the participants had two or more children (n = 615, 70.5%), almost all (n = 846, 97%) earned less than 500,000 Uganda Shillings ($140) per month. Table 1 provides details of the characteristics of the study participants stratified by health facility.

### Prevalence of appropriate perinatal care

The majority of the participant's care was classified as inappropriate care with 863 (99.0%) mothers receiving inappropriate care in the antenatal phase, 690 (79.1%) receiving

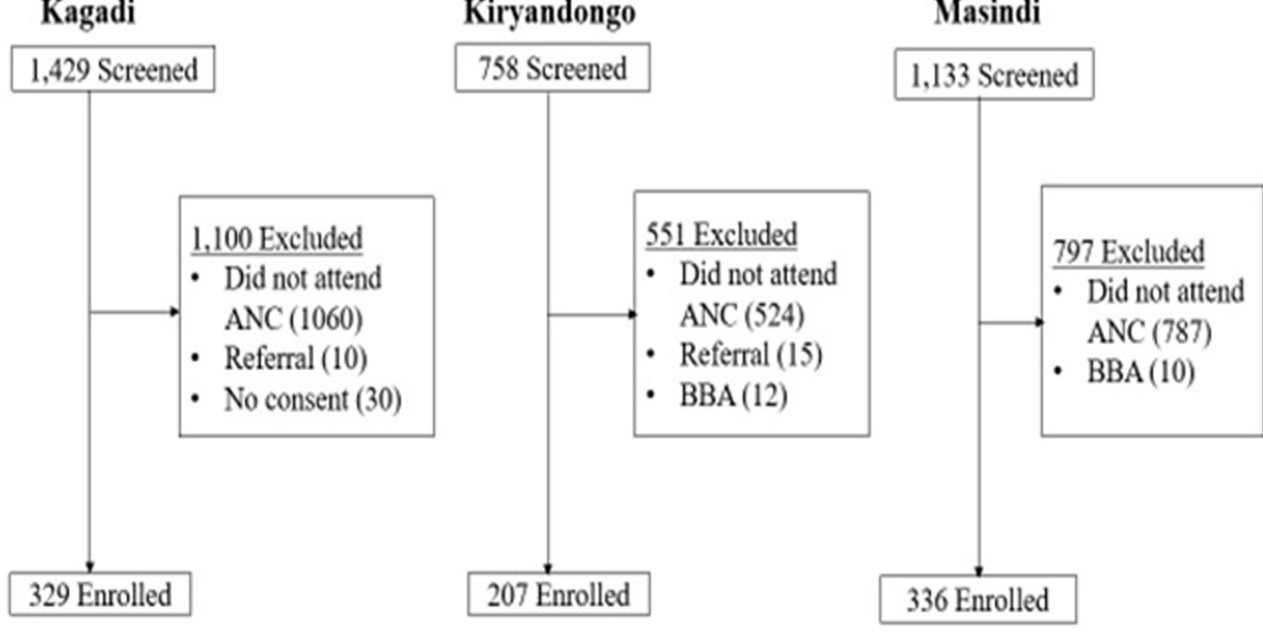

**Fig 1. Flowchart.** ANC = antenatal care; BBA = birth before arrival.

**Table 1. Characteristics of the study population.**

| Characteristic (n = 872) | Kagadi | Kiryandongo | Masindi | Total |
|---|---|---|---|---|
| | n (%) | n (%) | n (%) | n (%) |
| Number of participants | 329 | 207 | 336 | 872 |
| Age category of the mother | | | | |
| < 20 | 60 (18.2) | 46 (22.2) | 59 (17.6) | 165 (18.9) |
| 20–35 | 252 (76.6) | 148 (71.5) | 248 (73.8) | 648 (74.3) |
| >35 | 17 (5.2) | 13 (6.3) | 29 (8.6) | 59 (6.8) |
| Marital status | | | | |
| Married/Stable relationship | 290 (88.2) | 194 (93.7) | 298 (88.7) | 782 (89.7) |
| Single/divorced/separated | 39 (11.8) | 13 (6.3) | 38 (11.3) | 90 (10.3) |
| Education level | | | | |
| None/Primary | 187 (56.8) | 137 (66.2) | 129 (38.4) | 453 (52.0) |
| Secondary | 93 (28.3) | 65 (31.4) | 168 (50.0) | 326 (37.4) |
| Tertiary | 49 (14.9) | 5 (2.4) | 39 (11.6) | 93 (10.6) |
| Occupation | | | | |
| None | 16 (4.9) | 73 (35.3) | 209 (60.2) | 298 (34.2) |
| Informal employment | 280 (85.1) | 125 (60.4) | 89 (26.5) | 494 (56.6) |
| Formal employment | 33 (10.3) | 9 (4.3) | 38 (11.3) | 80 (9.2) |
| Income | | | | |
| <100, 000/ = | 75 (22.8) | 105 (50.7) | 240 (71.4) | 420 (48.2) |
| 100,000–500,000/ = | 237 (72.0) | 101 (48.8) | 88 (26.2) | 426 (48.8) |
| >500,000/ = | 17 (5.2) | 1 (0.5) | 8 (2.4) | 26 (3.0) |
| Parity | | | | |
| Primigravida (1) | 93 (28.3) | 61 (29.5) | 103 (30.7) | 257 (29.5) |
| Low parity (2–3) | 133 (40.4) | 76 (36.7) | 124 (36.9) | 333 (38.2) |
| Multipara (>3) | 103 (31.3) | 70 (33.8) | 109 (32.4) | 282 (32.3) |

inappropriate care in the intrapartum phase, and 795, (91.2%) receiving inappropriate care in the postpartum phase of care (Table 2). None of the participant's care in the antenatal or intrapartum period met the criteria of appropriate care. Appropriate care was only observed in the postpartum phase of care with 33/872 (3.8%) mothers' care meeting the WHO criteria for appropriate care. Kagadi hospital which contributed 329 (37.7%) of the study participants had no mother fulfilling the criteria for appropriate care in any of the three phases of care (Table 2). Masindi and Kiryandongo hospitals contributed almost equally to the proportion of participants that received appropriate care in the postpartum phase (51.5% versus 48.5% respectively p = 0.807). Of the 872 mothers enrolled, nine (1.0%) received no care in the antenatal phase, 182 (20.9%) received no care in the intrapartum phase, and 44 (5.1%) received no care in the postpartum phase of care.

Although overall prevalence of inappropriate care was high, the majority of the participants received at least 1 dose of tetanus toxoid (85%), received health education at least once during antenatal care (87.7%), had their labour monitored using a partograph (71.4%), had their partograph use initiated in the active first stage (78.3%), and had their vital signs assessed 24 hours after delivery (61.1%). The commonest components contributing to inappropriate antenatal care included: 1) mothers receiving less than the recommended eight antenatal visits (863, 99%); 2) initiating care after the 1st trimester (799, 91.6%); 3) missing the recommended physical exams (blood pressure, weight, checking for pallor, fetal heart rate, fundal height exam and checking for fetal lie and position) during the visits (858, 98.4%); 4) missing the

**Table 2. Prevalence of inappropriate perinatal care.**

| Characteristic | Kagadi | Kiryandongo | Masindi | All |
|---|---|---|---|---|
| | n (%) | n (%) | n (%) | n (%) |
| Number of participants (N) | 329 | 207 | 336 | 872 |
| Classification of Antenatal care received | | | | |
| No care | 0 (0.0) | 0 (0.0) | 9 (2.7) | 9 (1.0) |
| Inappropriate care | 329 (100) | 207 (100) | 327 (97.3) | 863 (99.0) |
| Appropriate | 0 (0.0) | 0 (0.0) | 0 (0.0) | 0 (0.0) |
| Classification of intrapartum care received | | | | |
| No care | 29(8.8) | 12 (5.8) | 141 (42.0) | 182 (20.9) |
| Inappropriate care | 300 (91.2) | 195 (94.2) | 195 (58.0) | 690 (79.1) |
| Appropriate care | 0 (0.0) | 0 (0.0) | 0 (0.0) | 0 (0.0) |
| Classification of Postpartum care received | | | | |
| No care | 21 (6.4) | 3 (1.5) | 20 (6.0) | 44 (5.1) |
| Inappropriate care | 308 (93.6) | 187 (90.3) | 300 (89.3) | 795 (91.2) |
| Appropriate care | 0 (0.0) | 17 (8.2) | 16 (4.7) | 33 (3.8) |

recommended laboratory tests during the antenatal contacts (872, 100%); and 5) no ultrasound scan before 24 weeks of amenorrhea (797, 91.4%) (Table 3). For labour and delivery, none of the mothers had the fetal heart, contractions, and pulse rate monitored every 30 minutes and only 23 (2.4%) mothers had cervical dilation, descent of presenting part, and maternal blood pressure monitored every 4 hours. Finally, the commonest factors contributing to inappropriate care in the postpartum phase included lack of an abdominal examination within 24 hours after delivery (807, 92.6%), and absence of vaginal examination/urine voiding assessment within 24 hours post-delivery (794, 91.1%).

## Factors associated with appropriate perinatal care

Factors significantly associated with appropriate care at multivariable analysis included maternal age and parity of the mother. The proportion of mothers having appropriate care increased with increasing maternal age (aPR = 2.7, 95% CI 0.9–8.5, p = 0.09 for mothers aged 20–35 years: and aPR = 11.9, 95% CI 2.8–51.4, p<0.001 for mothers >35 years). On the other hand, the proportion of mothers with appropriate care reduced with increasing parity. The prevalence of appropriate care was 70% lower in mothers with two-three children and those with more than three children compared to prime gravidas (aPR = 0.3 95% CI 0.1–0.9, p = 0.03, and aPR = 0.3 95% CI 0.1–0.8, p = 0.02 respectively). Mothers who sought care from Kiryandongo were more likely to receive appropriate care than those from Masindi although the association is of borderline significance (aPR = 1.9 95% CI 0.9–3.7, p = 0.06). Although the education level of the mother has previously been associated with the level of care they receive during the perinatal period [7, 27], the association was not significant in this study. Table 4 provides the details of the factors associated with appropriate postpartum care.

## Discussion

Improved perinatal care is essential in reducing maternal deaths and improving birth outcomes. In this study we described the prevalence and factors associated with appropriate perinatal care in Bunyoro region, Uganda. The study found that no participant received appropriate care in the antenatal and intrapartum periods of care, and only four in every one

**Table 3. Perinatal care received by mothers.**

| Characteristic | Kagadi | Kiryandongo | Masindi | All |
|---|---|---|---|---|
| | n (%) | n (%) | n (%) | n (%) |
| N | 329 | 207 | 336 | 872 |
| **Antenatal care Documentation** | | | | |
| Nature of ANC record | | | | |
| Health passport/ANC card | 107 (32.5) | 86 (41.6) | 93 (27.7) | 286 (32.8) |
| Exercise book/Papers | 222 (67.5) | 121 (58.4) | 243 (72.3) | 586 (67.2) |
| Completeness of ANC records | | | | |
| Complete | 24 (7.3) | 182 (87.9) | 202 (60.1) | 408 (46.8) |
| Incomplete | 305 (92.7) | 25 (12.1) | 134 (39.9) | 464 (53.2) |
| **Antenatal care visits** | | | | |
| Number of ANC contacts | | | | |
| < 8 contacts | 329 (100) | 205 (90.0) | 329 (97.9) | 863 (99) |
| 8 or more contacts | 0 (0.0) | 2 (1.0) | 7 (2.1) | 9 (1.0) |
| Initiation of Antenatal care | | | | |
| 1st trimester | 25 (7.6) | 21 (10.1) | 27 (8.0) | 73 (8.4) |
| After 1st trimester | 304 (92.4) | 186 (89.9) | 309 (92.0) | 799 (91.6) |
| **Levels of Antenatal care received** | | | | |
| Recommended exams conducted during every contact (Maternal status–BP, Wt, Pallor; Fetal status–FHR, Lie, & Position; FH) | | | | |
| No | 327 (99.4) | 197 (95.2) | 334 (99.4) | 858 (98.4) |
| Yes | 2 (0.6) | 10 (4.8) | 2 (0.6) | 14 (1.6) |
| Tests carried out during contacts as recommended (Hb, HIV, Syphilis, Urine glucose & Protein) | | | | |
| No | 329 (100) | 207 (100) | 336 (100) | 872 (100) |
| Yes | 0 (0.0) | 0 (0.0) | 0 (0.0) | 0 (0.0) |
| At least 1 Ultrasound scan done before 24 weeks | | | | |
| No | 279 (84.8) | 200 (96.6) | 318 (94.6) | 797 (91.4) |
| Yes | 50 (15.2) | 7 (3.4) | 18 (5.4) | 75 (8.6) |
| Preventive drugs given during contacts as recommended (Iron/Folic, Fansidar) | | | | |
| No | 223 (67.8) | 112 (54.1) | 111 (33.0) | 446 (51.2) |
| Yes | 106 (32.2) | 95 (45.9) | 225 (67.0) | 426 (48.9) |
| At least 1 Tetanus toxoid dose given at first contact | | | | |
| No | 42 (12.8) | 9 (4.3) | 80 (23.8) | 131 (15.0) |
| Yes | 287 (87.2) | 198 (95.7) | 256 (76.2) | 741 (85.0) |
| Health Education received at least once during any contact | | | | |
| No | 15 (4.6) | 3 (1.4) | 89 (26.5) | 107 (12.3) |
| Yes | 314 (95.4) | 204 (98.6) | 247 (73.5) | 765 (87.7) |
| **Labour and delivery care documentation** | | | | |
| Partograph use | | | | |
| No | 30 (9.1) | 13 (6.3) | 206 (61.3) | 249 (28.6) |
| Yes | 299 (90.9) | 194 (93.7) | 130 (38.7) | 623 (71.4) |
| Completeness of Partograph | | | | |
| Complete | 98 (32.8) | 143 (73.7) | 15 (11.5) | 256 (41.1) |
| Incomplete | 201 (67.2) | 51 (26.3) | 115 (88.5) | 367 (58.9) |
| Initiation of partograph use | | | | |
| Latent first stage (< 5cm) | 61 (20.4) | 47 (24.2) | 27 (20.8) | 135 (21.7) |
| Active first stage (5cm or more) | 238 (79.6) | 147 (75.8) | 103 (79.2) | 488 (78.3) |
| **Levels of labour and delivery care received** | | | | |

*(Continued)*

**Table 3.** (Continued)

| Characteristic | Kagadi | Kiryandongo | Masindi | All |
|---|---|---|---|---|
| | n (%) | n (%) | n (%) | n (%) |
| Recommended exams monitored every 4 hours during labour (Cervical dilatation, descent of the presenting part, maternal blood pressure) | | | | |
| No | 317 (96.4) | 196 (94.7) | 336 (100) | 849 (97.4) |
| Yes | 12 (3.6) | 11 (5.3) | 0 (0.0) | 23 (2.4) |
| Recommended exams monitored every 30 minutes during labour (Fetal heart rate, uterine contractions, maternal pulse rate) | | | | |
| No | 329 (100) | 207 (100) | 336 (100) | 872 (100) |
| Yes | 0 (0.0) | 0 (0.0) | 0 (0.0) | 0 (0.0) |
| **Levels of Postnatal care received** | | | | |
| Recommended Vital signs assessment within 24 hours after delivery (Blood pressure, pulse rate, and temperature) | | | | |
| No | 26 (7.9) | 17 (8.2) | 296 (88.1) | 339 (38.9) |
| Yes | 303 (92.1) | 190 (91.8) | 40 (11.9) | 533 (61.1) |
| Recommended abdominal examinations within 24 hours after delivery (Uterine contraction and fundal height) | | | | |
| No | 326 (99.1) | 190 (91.8) | 291 (86.6) | 807 (92.6) |
| Yes | 3 (0.9) | 17 (8.2) | 45 (13.4) | 65 (7.4) |
| Recommended vaginal bleeding and urine voiding assessment within 24 hours after delivery | | | | |
| No | 329 (100) | 173 (83.6) | 292 (86.9) | 794 (91.1) |
| Yes | 0 (0.0) | 34 (16.4) | 44 (13.1) | 78 (8.9) |

hundred participants received appropriate care in the postpartum period. Increasing maternal age and reducing parity were significantly associated with appropriate care.

Appropriate perinatal care in this setting is still limited and the study findings show that the performance of the region regarding the set minimum standards for perinatal care are still below the national averages [7]. In this study no mother received appropriate care in the antenatal and intrapartum period and only a few mothers received appropriate care in the postpartum period. Poor perinatal care increases the risk of poor maternal and birth outcomes [28], and the findings from this study could explain the high maternal mortality rates observed in the region [29]. Although health care providers schedule antenatal contacts for mothers, the time of initiating antenatal care and the total number of antenatal visits one has during a pregnancy are individual driven factors and represent the level of health care utilization in the region which has been shown to be consistently low [7, 22, 29], the other factors contributing to inappropriate care are health system driven suggesting poor health service delivery in the region. The incomplete/inappropriate records, missed physical examinations, and missed laboratory tests as observed in this study may lead to delays in appropriate decision making, timely interventions, and communication among the care providers which may cost both the lives of the mother and the baby.

The high levels of inappropriate care observed in the region may be attributed to shortage of resources like medical supplies, equipment, drugs, and staffing in hospitals as well as differences in competences of the health care providers, which are common occurrences in low income countries [30, 31]. The degree of this shortage and difference in competences may vary from facility to facility which creates a difference in care provision. However, in this setting several projects have been established to address the gaps in care due to shortage of resources by establishing effective, low-cost interventions. As part of the SMGL initiative and in collaboration with the Uganda Ministry of Health, the region received upgrade for number of public and private facilities to provide clean and safe basic delivery services and had its supply chains

**Table 4. Factors associated with appropriate postpartum care.**

| Characteristic | | n/N (%) | Unadjusted PR (95% CI) | P-value | Adjusted PR (95% CI) | P-value |
|---|---|---|---|---|---|---|
| Age category of the mother | < 20 | 5/165 (3.0) | 1.0 | | 1.0 | |
| | 20–35 | 21/648 (3.2) | 1.1 (0.4–2.8) | 0.891 | 2.7 (0.9–8.5) | 0.085 |
| | >35 | 7/59 (11.9) | 3.9 (1.3–11.9) | 0.016 | 11.9 (2.8–51.4) | 0.001 |
| Marital status | Married/Stable relationship | 32/782 (4.1) | 1.0 | | | |
| | Single/divorced/separated | 1/90 (1.1) | 0.3 (0.04–1.9) | 0.197 | | |
| Education level | None/primary | 19/453 (4.2) | 1.00 | | 1.00 | |
| | Secondary | 11/326 (3.4) | 0.8 (0.4–1.7) | 0.559 | 0.7 (0.3–1.5) | 0.325 |
| | Tertiary | 3/93 (3.2) | 0.8 (0.2–2.6) | 0.667 | 0.7 (0.2–3.1) | 0.681 |
| Occupation | None | 17/298 (5.7) | 1.0 | | | |
| | Informal employment | 13/494 (2.6) | 0.5 (0.9–1.1) | 0.032 | | |
| | Formal employment | 3/80 (3.8) | 0.7 (0.9–1.1) | 0.494 | | |
| Income | <100, 000/ = | 20/420 (4.8) | 1.0 | | 1.0 | |
| | 100,000–500,000/ = | 12/426 (2.8) | 0.6 (0.3–1.2) | 0.143 | 0.9 (0.5–1.8) | 0.777 |
| | >500,000/ = | 1/26 (3.8) | 0.8 (0.1–5.8) | 0.832 | 1.6 (0.2–10.1) | 0.632 |
| Parity | Prime gravida (1)) | 13/257 (5.1) | 1.0 | | 1.0 | |
| | Low parity (2–3) | 8/333 (2.4) | 0.5 (1.0–1.1) | 0.092 | 0.3 (0.1–0.9) | 0.027 |
| | Multipara (>3) | 12/282 (4.3) | 0.8 (1.0–1.1) | 0.659 | 0.3 (0.1–0.8) | 0.018 |
| Health Facility | Masindi | 16/336 (4.8) | 1.0 | | 1.0 | |
| | Kiryandongo | 17/207 (8.2) | 1.7 (0.9–3.3) | 0.106 | 1.9 (0.9–3.7) | 0.062 |
| | Kagadi | 0/329 (0.0) | 0 | N/A | 0 | N/A |

for essential supplies and medicines strengthened [17]. These efforts were supplemented by the Clinton Health Access Initiative and the URMCHIP that have equipped the health facilities in the region to ably manage, deliver and sustain improvement in reproductive, maternal, new born, child and adolescent Health in the region [15, 16]. With these interventions it is anticipated that shortages of resources would be limited in the country. In other low income countries, use of clinical mentorship, training, facility quality improvement teams, and quality of care reviews have been used to improve the quality of perinatal care with positive results over the study periods [32–34]. Similarly, in Uganda, SMGL implemented facility infrastructure upgrades, procured essential medical equipment, supplies and medications, trained and mentored health care providers, recruited health care providers, and strengthened maternal and perinatal mortality surveillance in parts of this region. As a result, availability of facility electricity and water increased, use of partograph in monitoring labour increased, and the proportion of facilities with no stock outs also increased [35]. Sustainability of such interventions beyond the study periods maybe important to consider for continued improvement.

Increasing maternal age was significantly associated with appropriate care with a clear dose effect relationship observed. Maternal age has been linked to demand for services, with increased age being able to demand for better services than those with lower age [7]. Findings from this study could attest to this explanation. The good care provided to older women in this study may lead to reduced adverse birth outcomes that are common in this age group [36].

Reducing parity was significantly associated with appropriate care. Mothers below 20 years of age are perceived to be at risk of having pregnancy related complications [37–40], which explains why mothers of that age group in this study could have received appropriate care. As a result, primigravida mothers could have had improved maternal and birth outcomes.

Much as higher education and income are associated with better care, this study did not find them significantly associated with appropriate postpartum care. The difference in findings may be attributed to the fact that the study sites were public hospitals where care is provided free of charge and mothers do not have to pay for most of the services. In addition, mothers with higher education and income are likely to seek private services instead of general services that are often provided to most of the mothers in public hospitals [7]. This could result in having less complications during childbirth.

This study had some limitations. First, it was a cross sectional study that merely provided a snapshot of the care processes in these facilities. Conducting a follow up study with both observational and extracted data could have provided a better assessment of the quality of care received by mothers. However, the findings are still valid as they highlight gaps and make suggestions on what could be done differently, which provides a basis for further studies. Secondly, this study extracted the data on care from patient records. It is possible that information recorded did not exactly reflect the care provided to mothers during these periods of care. In addition, there is no data on direct observation of the care processes which would have explained or validated the data extracted from the patient records. The source of data also makes it hard to appreciate the in-depth factors affecting care including personal factors like beliefs, values, experiences, and ability to access services as well as determine the usage of the data in patient care. However, exit interviews were held with mothers and care providers to understand the nature of care provided. Thirdly, the study excluded women who were referred for childbirth from other facilities and did not attend antenatal care in the study hospitals. This could have reduced the study's ability to assess care provided to postpartum women by other facilities in this region. Therefore, findings of this study may not be entirely generalizable to Bunyoro region. Lastly, the study did not capture information on type of pregnancy and existing co-morbidities which could have influenced the nature of care provided to mothers.

## Conclusion

In summary, we found that antenatal, intrapartum and postpartum care provided to mothers was below the standard recommended by WHO and Ministry of Health in Bunyoro, Uganda. Increased age and reduced parity influenced the extent to which mothers were cared for during the postpartum period. Therefore building the capacity of healthcare providers on the recommended perinatal care; as well as provision of required supplies, equipment and drugs; and provision of appropriate infrastructure by government could motivate health care providers to provide quality care. Mothers also can be empowered through health education to demand for care. There is need to develop sustainable strategies that could produce lasting impact on the quality of care. Developing an appropriate tool for assessing quality of care in this setting may also help researchers in assessing quality of care in similar settings.

## Supporting information

**S1 Appendix.**
(DOC)

**S1 Dataset.**
(XLS)

**S2 Dataset.**
(XLS)

**S3 Dataset.**
(XLS)

## Acknowledgments

We are thankful to all mothers who agreed to participate in the study, the health facility and perinatal unit leadership that made access to women and their records possible, and the research assistants who ensured that quality data is obtained. Special appreciation is given to the study site supervisors who ensured that data is timely accessed and collected. Lastly, we appreciate the district health officers for allowing this study to be conducted in their districts.

## Author Contributions

**Conceptualization:** Mercy Muwema, Dan K. Kaye, Grace Edwards, Gorrette Nalwadda, Joanita Nangendo, Wilson Mwanja, Elizabeth N. Ekong, Joan N. Kalyango, Joaniter I. Nankabirwa.

**Formal analysis:** Mercy Muwema, Joanita Nangendo, Jaffer Okiring, Wilson Mwanja, Joaniter I. Nankabirwa.

**Funding acquisition:** Mercy Muwema, Wilson Mwanja.

**Investigation:** Mercy Muwema, Wilson Mwanja, Elizabeth N. Ekong.

**Methodology:** Mercy Muwema, Dan K. Kaye, Grace Edwards, Joanita Nangendo, Wilson Mwanja, Joan N. Kalyango, Joaniter I. Nankabirwa.

**Project administration:** Mercy Muwema, Wilson Mwanja.

**Resources:** Mercy Muwema.

**Software:** Wilson Mwanja.

**Supervision:** Dan K. Kaye, Grace Edwards, Gorrette Nalwadda, Joan N. Kalyango, Joaniter I. Nankabirwa.

**Validation:** Joan N. Kalyango, Joaniter I. Nankabirwa.

**Writing – original draft:** Mercy Muwema, Wilson Mwanja, Joaniter I. Nankabirwa.

**Writing – review & editing:** Mercy Muwema, Dan K. Kaye, Grace Edwards, Gorrette Nalwadda, Joanita Nangendo, Jaffer Okiring, Wilson Mwanja, Elizabeth N. Ekong, Joan N. Kalyango, Joaniter I. Nankabirwa.

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
