## [Decision Letter · Decision Letter 0]

4 Jan 2022

PONE-D-21-34017Perinatal care in Western Uganda: Prevalence and factors associated with appropriate care among women attending three District HospitalsPLOS ONE

Dear Dr. Muwema,

Thank you for submitting your manuscript to PLOS ONE. After careful consideration, we feel that it has merit but does not fully meet PLOS ONE’s publication criteria as it currently stands. Therefore, we invite you to submit a revised version of the manuscript that addresses the points raised during the review process.

We look forward to receiving your revised manuscript.

Kind regards,

Ammal Mokhtar Metwally, Ph.D (MD)

Academic Editor

PLOS ONE

Journal Requirements:

Additional Editor Comments (if provided):

As this study is focusing on the ” Perinatal care in Western Uganda: Prevalence and factors associated with appropriate care among women in three District Hospitals”, so it is essential to consider enriching your discussion of the best intervention practices to use for improving and providing the appropriate care from countries with similar context, e.g. Enhancing the value of women's reproductive rights through community based interventions and strengths of community and health facilities based interventions in improving women care seeking behaviors…. etc..

However, this should be considered in addition to the reviewers’ remarks

Reviewers' comments:

Reviewer's Responses to Questions

**Comments to the Author**

1. Is the manuscript technically sound, and do the data support the conclusions?

Reviewer #1: Yes

Reviewer #2: Yes

Reviewer #3: Yes

2. Has the statistical analysis been performed appropriately and rigorously? 

Reviewer #1: Yes

Reviewer #2: Yes

Reviewer #3: Yes

3. Have the authors made all data underlying the findings in their manuscript fully available?

Reviewer #1: Yes

Reviewer #2: Yes

Reviewer #3: Yes

4. Is the manuscript presented in an intelligible fashion and written in standard English?

Reviewer #1: Yes

Reviewer #2: Yes

Reviewer #3: Yes

5. Review Comments to the Author

Reviewer #1: Overall, this paper was well presented. However, I struggled with reviewing the paper since the survey was not provided and detailed information on the WHO criteria was not included. This information should be included as part of the material. There was also no discussion about what might be the reasons for differences between the hospitals. Also, did the authors consider personal factors? Might there be some personal factors that may have impacted the results? Was this considered in the survey? For example, mother's beliefs, values, experiences, ability to access services (transport), spouse etc.?

Reviewer #2: Give greater attention to results in Table 3. They do provide some good news. For example, 49% of women got at least one preventive drug, 85% got tetanus toxoid, 88% got some health education, 71% had a partograph (although 59% were incomplete), 61% of those delivering had vital signs taken. By just citing aggregate scores in the summary, the article implies that women weren’t getting much of any services. You say that none of the mothers received appropriate antenatal or intrapartum care. True by your aggregate scores but some mother did receive some services. A professional health care provider might feel that you were saying that they weren’t doing anything!

What are the implications of 53% of ANC records incomplete. Could this lead to somewhat of an undercount of services or otherwise cast questions on results?

The article frequently uses the term quality of care. Actually it is more in Quantity of care (presence or absence of a task). For example, one indicator is whether a urine test is taken. But if it is, that is just the beginning. Was it tested? Correctly? Results communicated to the right people? Any relevant action taken?

Same with partograph— maybe it was drawn but did anyone use it to make important decisions? Some health practices may be more like rituals than action items…

No innovative strategies are mentioned although they are referred to several times…

This article indirectly raises the question of whether the WHO items are totally appropriate for rural Uganda? Are 8 perinatal visits essential? Which of the items are the most important? It would also be interesting to see if any of the WHO maternal care staff or advisors could provide all these services themselves in a setting like rural Uganda…."? I also wondered if Ugandan staff were providing other services (such as comforting women, food, water) that don’t appear in the list of items which are all health technologies rather than tender loving care. You might guess that I am a social scientist/public health person rather than a clinician…

We now refer to patients declining consent rather than refusing it.. it is their right to decline…

Was the 2016 Dhs the most recent one? Misspelling in Footnote 20.

A nice, well written paper and amazing collaboration… I suppose the article would be useful as a very rough baseline to for studying local improvements over the years. But if I were doing this work, I would seek feedback from local health staff about which items they thought were most practical and most likely to improve the health of mothers and babies and then drill down hard on these tasks. Cheers!

Reviewer #3: Well-done on your submission. This is a much needed study on an important topic. Please see below my comments:

Introduction

1) In the first paragraph of the introduction, there is no need to list out SDG 3, this is easily accessible. A reference would do.

2) In the second paragraph of the introduction, please provide a reference for this statement "Uganda is one of the countries that face challenges in provision of quality perinatal care".

3) In the second paragraph of the introduction, you wrote "According to the latest Uganda Demographic and Health Survey (DHS),......". Which is the latest Uganda DHS?

4) The introduction does not contain enough information to validate the rationale for the study. The introduction should provide a snapshot of the available knowledge on the topic and highlight the gap in the existing knowledge which this present study seeks to address. I would recommend a proper literature review of the prevalence and factors associated with perinatal care should be done. This will provide a better buildup to the study rationale.

Discussion

5) The authors confirmed that appropriate perinatal care was limited in the study setting and that "the performance of the region regarding the set minimum standards for perinatal care are still below the national averages". Then what is the basis of the study results when the study setting does not meet the set minimum standards? It would have made more sense if the study was on factors associated with (in)appropriate care among women in three District Hospitals. I am of the opinion that the study findings is skewed by the fact that the study setting does not meet the set minimum standards for perinatal care.

6) What are the strengths of the study?

7) What are the policy implications of the study?

6. PLOS authors have the option to publish the peer review history of their article (what does this mean?). If published, this will include your full peer review and any attached files.

Reviewer #1: No

Reviewer #2: No

Reviewer #3: No

---

## [Author Response · Author response to Decision Letter 0]

17 Feb 2022

REVIEWER: 1

GENERAL COMMENTS

GENERAL COMMENT 1:

Overall, this paper was well presented. However, I struggled with reviewing the paper since the survey was not provided and detailed information on the WHO criteria was not included. This information should be included as part of the material

RESPONSE: 

Thank you for pointing this out to us. The survey questionnaire had two sections, with the first section covering the biodata for the women and the second section eliciting the care received by the women during antenatal, intrapartum, and postpartum periods as seen in their patient records. The study used three forms of WHO recommendations: 1) recommendations on antenatal care for a positive pregnancy experience which describes the recommended care for women during pregnancy, how it should be given, 2) intrapartum care for a positive childbirth experience which provides the standard care to be given to women during labour and childbirth, and 3) recommendations on postnatal care of the mother and newborn that describe the standard care to be given to women and their babies during the six weeks after childbirth. The questionnaire and all the WHO recommendations used in the study have been included among the materials uploaded with the revised manuscript.

GENERAL COMMENT 2:

There was also no discussion about what might be the reasons for the differences between the hospitals.

RESPONSE:

The differences observed in the quality of care in the different hospitals may be due to the differences in staffing levels and competences and availability of medicines, supplies, and equipment. The manuscript discussion has been revised to include the possible reasons for the differences between hospitals (Page 20 Line 289-292).

GENERAL COMMENT 3:

Also, did the authors consider personal factors? Might there be some personal factors that may have impacted the results? Was this considered in the survey? For example, mother's beliefs, values, experiences, ability to access services (transport), spouse etc.?

RESPONSE:

The study used medical records as the primary source of data to answer the study objectives. As part of the data collection, participant demographics including age, marital status, education, occupation, income, and parity were collected and have been included in the factors assessed. Unfortunately, based on the source of data, we were unable to collect in-depth information on other personal factors like beliefs, values, experiences and ability to access services. This has been included as part of the limitations to the study (Page 22, lines 335-337). 

REVIEWER: 2

SPECIFIC COMMENTS

SPECIFIC COMMENT 1:

Give greater attention to results in Table 3. They do provide some good news. For example, 49% of women got at least one preventive drug, 85% got tetanus toxoid, 88% got some health education, 71% had a partograph (although 59% were incomplete), 61% of those delivering had vital signs taken. By just citing aggregate scores in the summary, the article implies that women weren’t getting much of any services. You say that none of the mothers received appropriate antenatal or intrapartum care. True by your aggregate scores but some mother did receive some services. A professional health care provider might feel that you were saying that they weren’t doing anything!

RESPONSE

Yes, it is true that women received some care. Majority of the participants received at least 1 dose of tetanus toxoid (85%), received health education at least once during antenatal care (87.7%), had their labour monitored using a partograph (71.4%), had their partograph use initiated in the active first stage (78.3%), and had their vital signs assessed 24 hours after delivery (61.1%). However composite variables for antenatal, intrapartum and postpartum constituting of various care components as per the WHO recommendations were created. A response of “yes” was given to mean that care was provided in its totality. Care provided in partiality to the mentioned recommendations was regarded as “no.” The manuscript has been revised to include findings of some care received by women (Page 13, line 224 - 228). 

GENERAL COMMENTS

GENERAL COMMENT 1:

What are the implications of 53% of ANC records incomplete? Could this lead to somewhat of an undercount of services or otherwise cast questions on results?

RESPONSE

 Yes, we agree that the high percentage of incomplete records could have led to an undercount of services provided. We have acknowledged this as part of the limitation of the study (page 21, line 332-333).

GENERAL COMMENT 2

The article frequently uses the term quality of care. Actually it is more in Quantity of care (presence or absence of a task). For example, one indicator is whether a urine test is taken. But if it is, that is just the beginning. Was it tested? Correctly? Results communicated to the right people? Any relevant action taken?

Same with partograph— maybe it was drawn but did anyone use it to make important decisions? Some health practices may be more like rituals than action items…

RESPONSE

It is true that the term quality of care goes beyond presence or absence of a task to include results and how the results influenced care. It is for this reason that the study derived evidence of care provided from the women hand held cards/passport/files and not the facility registers. The women files indicated whether care had been provided, when it was provided, and the results in case of a test. When care was not provided, it was not documented at all. Therefore a response of “yes” was given to mean that care was provided and results were recorded in case of a test. It was assumed that once care is recorded in that particular woman’s record, then it has been communicated to her and has influenced the care decisions made. The study could however not rule out errors in documentation neither could it ascertain the accuracy of the test taken since care was being studied retrospectively in units of care and not the laboratory. The study scope did not include action taken or decisions taken much as some were documented. We have acknowledged this as part of the limitation of the study (page 22, line 337-338). 

GENERAL COMMENT 3

No innovative strategies are mentioned although they are referred to several times…

RESPONSE:

Innovative strategies like clinical mentorship, training, use of facility quality improvement teams, and quality of care reviews have been used to improve care received by women during the perinatal period in developing countries. The discussion section of the manuscript has been revised to include these best practices (Page 20 line 301-310).

GENERAL COMMENT 4

This article indirectly raises the question of whether the WHO items are totally appropriate for rural Uganda. Are 8 perinatal visits essential? Which of the items are the most important? It would also be interesting to see if any of the WHO maternal care staff or advisors could provide all these services themselves in a setting like rural Uganda…."? 

RESPONSE:

We agree that the WHO questionnaire may not yet be appropriate tool for assessing care in resource limited settings like Uganda, however, given that; 1) the recommendations for care in the country are based on the WHO guidelines; 2) the country has not developed any other tools for assessing care standards; 3) other studies have used the same tool for assessing care and using it allows for comparison, we believe the findings are still valid to inform practice and improvement in care. 

GENERAL COMMENT 5

I also wondered if Ugandan staff were providing other services (such as comforting women, food, water) that don’t appear in the list of items which are all health technologies rather than tender loving care. You might guess that I am a social scientist/public health person rather than a clinician…

RESPONSE:

The recommendations for care in the country are based on the WHO guidelines and the country has not developed any other additional guidelines in regard to provision of routine care to women during the perinatal period. The recommendations for care do not provide for comforting women nor provision of food and water to the women by the health care providers.

GENERAL COMMENT 6

We now refer to patients declining consent rather than refusing it. It is their right to decline…

RESPONSE:

Thank you for the correction. The manuscript has been edited and area with this phrase has been replaced with “declining consent” (Page 10 Line 191).

GENERAL COMMENT 7

Was the 2016 DHS the most recent one? Misspelling in Footnote 20.

RESPONSE:

Yes, 2016 demographic health survey (DHS) is the most recent survey in Uganda. Reference 23 (Page 25, line 434-436) has been edited to remove the misspelling in the term “services.”

GENERAL COMMENT 8

A nice, well written paper and amazing collaboration… I suppose the article would be useful as a very rough baseline to for studying local improvements over the years. But if I were doing this work, I would seek feedback from local health staff about which items they thought were most practical and most likely to improve the health of mothers and babies and then drill down hard on these tasks. Cheers!

RESPONSE:

Thank you for this valuable recommendation. The research team agrees with you and has already considered this very important question in their next study on developing an improved care delivery model for women during the perinatal period. 

REVIEWER: 3

SPECIFIC COMMENTS

SPECIFIC COMMENT 1

In the first paragraph of the introduction, there is no need to list out SDG 3, this is easily accessible. A reference would do.

RESPONSE:

Thank you for this comment. We have discussed the benefits and risks of having the 3rd SDG and agree that although referenced, some of the audience may not have the time to check reference and since it does not alter the message, we have decided to maintain it. 

SPECIFIC COMMENT 2

In the second paragraph of the introduction, please provide a reference for this statement "Uganda is one of the countries that face challenges in provision of quality perinatal care.”

RESPONSE:

Thank you for your comment. The reference to this is reference 7 to 11. The manuscript has been edited to include the references (Page 4, line 65). 

SPECIFIC COMMENT 3 

In the second paragraph of the introduction, you wrote "According to the latest Uganda Demographic and Health Survey (DHS),......" Which is the latest Uganda DHS?

RESPONSE:

The manuscript has been edited to include the year “2016” as the latest Uganda DHS (page 4, line 65)

SPECIFIC COMMENT 4 

The introduction does not contain enough information to validate the rationale for the study. The introduction should provide a snapshot of the available knowledge on the topic and highlight the gap in the existing knowledge which this present study seeks to address. I would recommend a proper literature review of the prevalence and factors associated with perinatal care should be done. This will provide a better buildup to the study rationale.

RESPONSE:

Thank you for the recommendation. There is limited literature on the prevalence and factors associated with perinatal care in Uganda, however, we have made attempts revising the manuscript to include the literature we have come across (pg 4, line 75-76).

SPECIFIC COMMENT 5

What are the strengths of the study?

RESPONSE:

The biggest strength of the study was that it highlights gaps and makes suggestions on what could be done differently, which provides a basis for further studies. The manuscript has been revised to indicate the strength of the study (pg 21, line 330-331).

SPECIFIC COMMENT 6

What are the policy implications of the study?

RESPONSE:

The findings have significant policy implications including; 1) The need to ensure availability of trainings for health care providers, supplies, medicines, support supervision, and appropriate infrastructure to improve care, 2) the need to develop an appropriate tool for assessment of quality care in this setting, and 3) the need to develop sustainable strategies that may produce lasting impact on quality of care. The manuscript has been revised to include the policy implications of the study (pg 22, line 354-356).

GENERAL COMMENTS

GENERAL COMMENT 1

The authors confirmed that appropriate perinatal care was limited in the study setting and that "the performance of the region regarding the set minimum standards for perinatal care are still below the national averages". Then what is the basis of the study results when the study setting does not meet the set minimum standards? It would have made more sense if the study was on factors associated with (in)appropriate care among women in three District Hospitals. I am of the opinion that the study findings is skewed by the fact that the study setting does not meet the set minimum standards for perinatal care.

RESPONSE

The importance of the study was to: 1) To highlight the challenges of care provision in the region, 2) make the challenges in care provision known to the policy makers and other stakeholders, and 3) support the design of an improved and appropriate care package for the setting using the study results. 

ADDITIONAL COMMENT:

As this study is focusing on the ” Perinatal care in Western Uganda: Prevalence and factors associated with appropriate care among women in three District Hospitals”, so it is essential to consider enriching your discussion of the best intervention practices to use for improving and providing the appropriate care from countries with similar context, e.g. Enhancing the value of women's reproductive rights through community based interventions and strengths of community and health facilities based interventions in improving women care seeking behaviors…. etc..

RESPONSE:

Thank you for this thoughtful comment. The discussion section of the manuscript has been revised to include the best practices that have been used to improve care received by women during the perinatal period in developing countries (Page 20, line 301-310).

---

## [Decision Letter · Decision Letter 1]

1 Apr 2022

Perinatal care in Western Uganda: Prevalence and factors associated with appropriate care among women attending three District Hospitals

PONE-D-21-34017R1

Dear Dr. Muwema,

We’re pleased to inform you that your manuscript has been judged scientifically suitable for publication and will be formally accepted for publication once it meets all outstanding technical requirements.

Kind regards,

Ammal Mokhtar Metwally, Ph.D (MD)

Academic Editor

PLOS ONE

Additional Editor Comments (optional):

A great effort was made by the authors to utilize the feedback that was provided for them to correct for resubmission

Reviewers' comments:

Reviewer's Responses to Questions

**Comments to the Author**

1. If the authors have adequately addressed your comments raised in a previous round of review and you feel that this manuscript is now acceptable for publication, you may indicate that here to bypass the “Comments to the Author” section, enter your conflict of interest statement in the “Confidential to Editor” section, and submit your "Accept" recommendation.

Reviewer #2: All comments have been addressed

2. Is the manuscript technically sound, and do the data support the conclusions?

Reviewer #2: Yes

3. Has the statistical analysis been performed appropriately and rigorously? 

Reviewer #2: Yes

4. Have the authors made all data underlying the findings in their manuscript fully available?

Reviewer #2: Yes

5. Is the manuscript presented in an intelligible fashion and written in standard English?

Reviewer #2: Yes

6. Review Comments to the Author

Reviewer #2: I believe that this paper will be useful for future researchers looking at the same topic in Uganda or similar

countries. It demonstrates the pitfalls of using uncritically WHO standards in a rural setting in Uganda and not delving into quality and appropriateness of care.

It points out the need for Ugandan standards of care appropriate for more rural settings. Ultimately, one needs to know which elements of care are given, which ones are relevant and have impact on mortality and morbidity. As such, it is a very first step from which others can learn.

Of course, women coming to hospitals may not be typical of the broader population. For example, there may be more emergency cases and possibly better off women who can afford care.

It shows how challenging this kind of research is. The authors responded well to the suggestions of the reviewers. Dont get discouraged! It is important research.

7. PLOS authors have the option to publish the peer review history of their article (what does this mean?). If published, this will include your full peer review and any attached files.

Reviewer #2: **Yes: **NANCY E WILLIAMSON

---

## [Editor Report · Acceptance letter]

13 May 2022

PONE-D-21-34017R1 

Perinatal care in Western Uganda: Prevalence and factors associated with appropriate care among women attending three District Hospitals 

Dear Dr. Muwema:

I'm pleased to inform you that your manuscript has been deemed suitable for publication in PLOS ONE. Congratulations! Your manuscript is now with our production department. 

Kind regards, 

on behalf of

Professor Ammal Mokhtar Metwally 

Academic Editor

PLOS ONE